# Learning semantic similarity in a continuous space

**Michel Deudon**
Ecole Polytechnique
Palaiseau, France
`michel.deudon@polytechnique.edu`

## Abstract

We address the problem of learning semantic representation of questions to measure similarity between pairs as a continuous distance metric. Our work naturally extends Word Mover's Distance (WMD) [1] by representing text documents as normal distributions instead of bags of embedded words. Our learned metric measures the dissimilarity between two questions as the minimum amount of distance the *intent* (hidden representation) of one question needs to "travel" to match the intent of another question. We first learn to *repeat, reformulate* questions to infer *intents* as normal distributions with a deep generative model [2] (variational auto encoder). Semantic similarity between pairs is then learned discriminatively as an optimal transport distance metric (Wasserstein 2) with our novel *variational siamese* framework. Among known models that can read sentences individually, our proposed framework achieves competitive results on Quora duplicate questions dataset. Our work sheds light on how deep generative models can approximate distributions (semantic representations) to effectively measure semantic similarity with meaningful distance metrics from Information Theory.

## 1    Introduction

Semantics is the study of meaning in language, used for understanding human expressions. It deals with the relationship between signifiers and what they stand for, their denotation. Measuring semantic similarity between pairs of sentences is an important problem in Natural Language Processing (NLP), for conversation systems (chatbots, FAQ), knowledge deduplication [3] or image captioning evaluation metrics [4] for example. In this paper, we consider the problem of determining the degree of similarity between pairs of sentences. Without loss of generality, we consider the case where the prediction is binary (duplicate pair or not), a problem known as paraphrase detection or semantic question matching in NLP.

A major breakthrough in the field of semantics was the work done in [5] and [6]. The authors proposed unsupervised methods to learn semantic representations of words (Word2vec [5], GloVe [6]). These representations come in the form of vectors (typically of dimension 50 - 300) that capture relationships between words in a geometric space. For instance, let $\nu_m \in \mathbb{R}^d$ denote the learned representation for word $m$. The following property holds $\nu_{king} - \nu_{man} + \nu_{woman} \simeq \nu_{queen}$ (linear substructure). In [1], the authors proposed a novel similarity metric for text documents: Word Mover's Distance (WMD). WMD [1] measures the dissimilarity between two bag-of-vectors (embedded words) with an optimal transport distance metric, Wasserstein 1, also known as Earth Mover's Distance [7] [8] [9]. The proposed method is conceptually appealing and accurate for news categorization and clustering tasks. Yet, WMD experimentally struggles to finely capture the semantics of shorter pairs (sentences, questions) because of the bag-of-word assumption. Indeed,*"why do cats like mice ?"* and *"why do mice like cats ?"* are mapped to the same bag-of-vectors representation, thus classified as duplicate. WMD does not capture the order in which words appear. Furthermore, because a computationally expensive discrete transportation problem is required to solve WMD, its use in conversation systems

and scalable online knowledge platforms is limited. A solution to the computational cost of WMD is to map a sentence to a single vector (instead of a bag) and then measure similarity between pairs as a Euclidean distance or cosine similarity in the mapped vectorial space. This way, there is no need for a transportation scheme to transform a document in another one. A simple approach to obtain a sentence representation from words' embeddings is to compute the barycenter of embedded words. Variants consider weighted sums (TF-IDF [10], Okapi BM-25 [11], SIF [12]) and top k principal components removal [12] [13]. However, none of these models finely capture semantics, again because of the bag-of-word assumption.

Convolutional Neural Networks (CNN) [14] [15], Recurrent Neural Networks (RNN) [16] [17] and Self-Attentive Neural Networks [18] [19] have been successfully used to model sequences in a variety of NLP tasks (sentiment analysis, question answering, machine translation to name a few). For semantic question matching (on the Quora dataset) and natural language inference (on Stanford and Multi-Genre NLI datasets [20] [21]), the state-of-the-art approach, Densely Interactive Inference Network (DIIN) [22], is a neural network that encodes sentences jointly rather than separately. Reading sentences separately makes the task of predicting similarity or entailment more difficult, but comes with a lower computational cost and higher throughput, since individual representations can be precomputed for fast information retrieval for instance. Models that can read sentences separately are conceptually appealing since they come with individual representations of sentences for visualization (PCA, t-SNE, MDS) and transfer to downstream tasks. Siamese networks are a type of neural networks that appeared in vision (face recognition [23]) and have recently been extensively studied to learn representations of sentences and predict similarity or entailment between pairs as an end-to-end differentiable task [3] [4] [24] [25].

In our work, we replace the discrete combinatorial problem of Word Mover's Distance (Wasserstein 1) [1] by a continuous one between normal distributions (intents), as illustrated in Figure 1. We first learn to *repeat, reformulate* with a deep generative model [2] (variational auto encoder) to learn and infer intents (step 1). We then learn and measure semantic similarity between question pairs with our novel *variational siamese* framework (step 2), which differs from the original one ([3] [4] [23] [24] [25]) as our hidden representations consist of two Gaussian distributions instead of two vectors. For intents with diagonal covariance matrices, the expression of Wasserstein 2 is explicit and computationally cheap. Our approach is evaluated on Quora duplicate questions dataset and performs strongly.

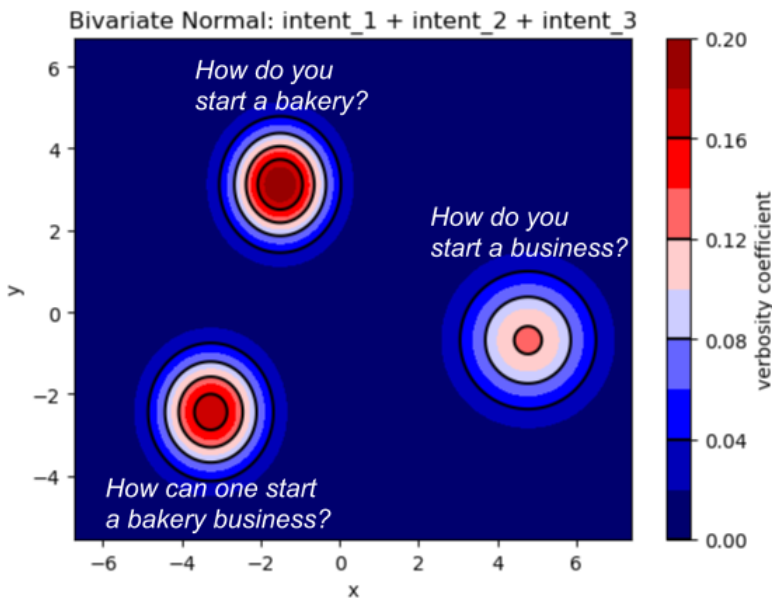

Figure 1: Illustration of our learned latent space. Sentences are mapped to Gaussian distributions $N(\mu, \sigma)$ (intents). The plot was obtained with PCA on the sentences' mean vectors $\mu$ after training of our *variational siamese* network. The covariance matrices $\sigma^2$ are diagonal.

In Section 2 we will provide background and context. Then, in Section 3 we will present our method to learn semantic representation of sentences. In Section 4 we will discuss how we measure similarity in our latent space. Finally, we will describe experiments, and present results in Section 5 before concluding.

## 2 Related work

### 2.1 Embedding words

An important principle used to compute semantic representations of sentences is the principle of compositionality which states that the meaning of a phrase is uniquely determined by the meaning of its parts and the rules that connect those parts. Word2Vec [5] and GloVe [6] are semantic embeddings of words based on their context and frequency of co-occurence in a text corpus such as Wikipedia, Google News or Common-Crawl. The key idea behind Word2Vec and Glove is that similar words appear in similar contexts. Word2Vec [5] learns semantic representation of words by learning to predict a word given its surrounding words (CBOW model) or vice versa (skip-gram model). GloVe [5] operates on a global word-word co-occurence matrix, learning a low-rank approximation for it. Word2Vec and Glove vectors effectively capture semantic similarity at the word level. [26] and [27] proposed embedding words in hyperbolic spaces to account for hierarchies in natural language (Zipf law). [28] proposed Word2Gauss to model uncertainty in words and naturally express assymetries for similarity/entailment tasks, using KL divergence.

### 2.2 Learning sentences' representations

Composing word vectors allows to encode sentences into fixed-size representations. The following presents how unsupervised neural auto encoders and variants can learn meaningful sentence representations, given a sequence of embedded words.

**Auto encoders and variants**   Neural autoencoders are models where output units are identical to input units [29]. An input $s$ is compressed with a neural encoder into a representation $z = q_\phi(s)$, which is then used to reconstruct $s$. Sequential Denoising Autoencoder (SDAE) [30] introduces noise in the input to predict the original source sentence $s$ given a corrupted version of it $s'$. The noise function independently deletes input words and swaps pairs with some probability. The model then uses a recurrent neural network (LSTM-based architecture [31]) to recover the original sentence from the corrupted version. The model can then be used to encode new sentences into vector representations. SDAEs are the top performer on paraphrase identification, among unsupervised model [30]. Another approach is Skip Thought (ST) vectors [32] which adapt the skip-gram model for words to the sentence level, by encoding a sentence to predict the sentences around it. Consequently, ST vectors require a consequent training corpus of ordered sentences, with a coherent narrative. On the SICK sentence relatedness benchmark [33], FastSent, a bag-of-word variant of Skip Thought, performs best among unsupervised model [30].

**Variational auto encoders**   Rather than representing sentences with fixed points $\mu \in \mathbb{R}^d$ in a d-dimensional space, an alternative is to represent them with normal distributions $\mu, \sigma \in \mathbb{R}^d$, as shown in Figure 1. Intuitively, the variance term accounts for uncertainty or ambiguity, a desirable property for modeling language. Think of the sentences *"He fed her cat food."* or *"Look at the dog with one eye."* for example. Furthermore, thanks to the variance term, the learned representations smoothly "fill" the semantic space to generalize better. Normal representation of sentences can be learned with Variational Auto Encoders (VAE), a class of deep generative models first proposed in [34] [35]. Similar to autoencoders, VAE learn to reconstruct their original inputs from a latent code $z$. However, instead of computing deterministic representations $z = q_\phi(s)$, the VAE learns, for any input $s$, a posterior distribution $q_\phi(z|s)$ over latent codes $z$. In terms of semantics, the latent distributions grant the ability to sum over all the possibilities (different meanings, different interpretations).

Formally, the VAE consists of an encoder $\phi$ and a decoder $\theta$. The VAE's encoder parameterizes the posterior distribution $q_\phi(z|s)$ over the latent code $z \in R^h$, given an input $s$. The posterior $q_\phi(z|s)$ is usually assumed to be a Gaussian distribution:

$$z \sim N(\mu(s), \sigma(s)) \tag{1}$$

where the functions $\mu(s), \sigma(s)$ are nonlinear transformations of the input s. The VAE's decoder parameterizes another distribution $p_\theta(s|z)$ that takes as input a random latent code z and produces an observation s. The parameters defining the VAE are learned by maximizing the following lower bound on the model evidence $p(s|\theta, \phi)$:

$$L_{\theta;\phi}(s) = E_{q_\phi(z|s)}[\log p_\theta(s|z)] - KL(q_\phi(z|s)||p(z)) \qquad (2)$$

where p(z) is a prior on the latent codes, typically a standard normal distribution $N(0, I)$.

To encode sentences into fixed-size representation, [36] proposed a VAE conditioned on a bag-of-word description of the text input. [2] employed long short-term memory (LSTM) networks [31] to read (encoder $\phi$) and generate (decoder $\theta$) sentences sequentially. The authors [2] proposed KL annealing and dropout of the decoder's inputs during training to circumvent problems encountered when using the standard LSTM-VAE for the task of modeling text data.

## 2.3   Learning similarity metrics

Siamese networks can learn similarity metrics discriminatively as suggested in [23]. They consist of two encoders (sharing the same weights), that read separately pairs of inputs into fixed sized representations (vectors). The concatenation $[u, v]$ [4] and/or Hadamard product and squared difference of the hidden vectors $[uv, |u - v|^2]$ [24] is then used as input layer for a Multi-Layer Perceptron that learns and predicts similarity or entailment of the corresponding pair. Siamese LSTM [3] and Siamese GRU networks [25] operate on sentences and achieve good results on paraphrase detection and question matching. Our proposed *variational siamese* network extends standard siamese networks to handle Gaussian hidden representations. It replaces the discrete transportation problem of Word Mover's Distance [1] (Wasserstein 1) by a continuous one (learning Wasserstein 2).

# 3   Learning to repeat, reformulate

## 3.1   Infering and learning intents

Our intuition is that behind semantically equivalent sentences, the intent is identical. We consider that two duplicate questions or paraphrases were generated from a same latent intent (hidden variable). We thus take the VAE approach to learn and infer semantic representation of sentences, but instead of only recovering sentence s from s ($repeat$), we also let our model learn to recover semantically equivalent sentences s' from s ($reformulate$), as illustrated in Figure 2. This first task (repeat, reformulate) provides a sound (meaningful) initialization for our encoder's parameters $\phi$ (see Figure 2). We then learn semantic similarity discriminatively as an optimal transport metric in the intent's latent space (covered in section 4). We did not consider training our model jointly on both tasks (generative and discriminative), as studied in [37]. We suspect the normal prior $N(0, I)$ on the hidden intents in equations (2) and (4) to conflict with our final objective of predicting similarity or entailment (blurry boundaries). Our *repeat, reformulate* framework is in the spirit of the work proposed in [38] where the authors employ a RNN-VAE and a dataset of sequence-outcome pairs to continuously revise discrete sequences, for instance to improve sentence positivity. [39] proposed learning paraphrastic sentence embeddings based on supervision from the Paraphrase Database [40] and a margin based loss with negative samples. Our approach differs from [39] as our hidden intents are Gaussians.

## 3.2   Neural architecture

Let $s, s'$ be a pair of semantically equivalent sentences (same content), represented as two sequences of word vectors, $s = w_1, w_2, ...w_{|s|}, w_i \in \mathbb{R}^d$. Our goal is to predict $s'$ given $s$. A sentence $s$ is always semantically equivalent to itself. This means we could include any sentence $s$ in the learning process (semi-supervised learning). Following [2], our encoder $\phi$ consists of a single layer bi-LSTM [31] network that encodes $s$ in a fixed length vector $c(s) \in \mathbb{R}^{2h}$. We linearly transform $c(s)$ in a hidden mean vector $\mu(s) \in \mathbb{R}^h$ and a hidden log diagonal covariance matrix $log(\sigma(s)^2) \in \mathbb{R}^h$ that define a Gaussian distribution $N(\mu(s), \sigma(s))$. Using the reparameterization trick (for backpropagation of the gradients), we sample intents $z$ from the posterior:

$$z \sim \mu(s) + \sigma(s)N(0, I). \qquad (3)$$

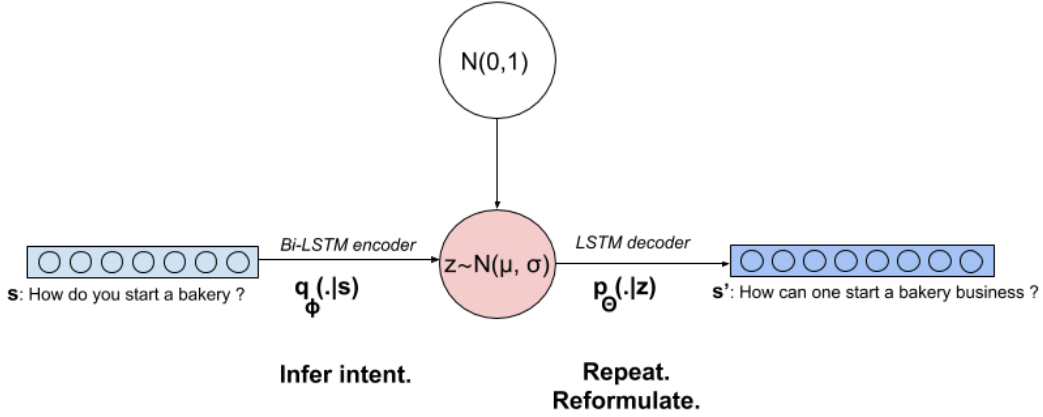

Figure 2: Our Bayesian framework and neural architecture to learn semantic representation of sentences, similar to [2].

Similar to [2], we then generate $s'$ with a single layer LSTM [31] decoder network $\theta$ conditioned on a sampled intent $z \in \mathbb{R}^h$. The true words of $s'$ are sequentially fed to the LSTM decoder during training (teacher-forcing). As in [2], we employ word dropout for the decoder with 60% keep rate. Our model learns semantic representation of sentences $N(\mu(s), \sigma(s))$ by learning to paraphrase. The parameters of our encoder-decoder architecture are learned by minimizing the following regularized loss by stochastic gradient descent:

$$-L_{\theta;\phi}(s, s') = -E_{q_\phi(z|s)}[\log p_\theta(s'|z)] + \kappa KL(q_\phi(z|s)||N(0, I)) \qquad (4)$$

where $\kappa = sigmoid(0.002(step - 2500))$ is a sigmoid annealing scheme, as suggested in [2]. The first term encourages sampled intents to encode useful information to repeat or reformulate questions. The second term enforces the hidden distributions (intents) to match a prior (a standard normal) to fill the semantic space ($\sigma > 0$) and smoothly measure similarity as an optimal transport distance metric.

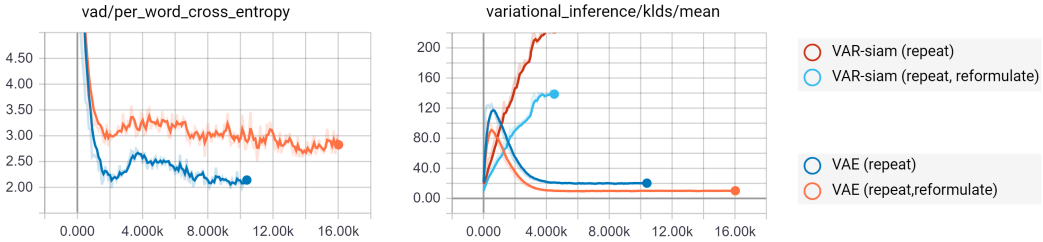

Figure 3: Results from our generative pretraining: $repeat$ (dark blue), $repeat, reformulate$ (orange). Left: Per word cross entropy. Right: KL divergence $KL(q_\phi(z|s)||N(0, I))$.

## 4 Variational siamese network

### 4.1 Learning semantic similarity in a latent continuous space

We train our model successively on two tasks (generative and discriminative) and use the first step as a smooth initialization ($N(0, I)$ prior and semi-supervised setting) for the second one, learning similarity. Our work differs from [3] [4] [23] [24] [25] as our encoded states (intents $z_s$) are multivariate Gaussians with diagonal covariance matrices $N(\mu, \sigma)$. This allows us to smoothly measure semantic similarity with different distance metrics or divergences from Information Theory as in [28][41]. The computation of Wasserstein 2 ($W_2^2$) and Mahalanobis distance ($D_M$) is efficient for two Gaussians with diagonal covariance matrices $p_1 = N(\mu_1, \sigma_1)$ and $p_2 = N(\mu_2, \sigma_2)$ on $\mathbb{R}^h$.

$$W_2^2(p_1, p_2) = \sum_{i=1}^{h} (\mu_1^i - \mu_2^i)^2 + (\sigma_1^i - \sigma_2^i)^2 \tag{5}$$

$$D_M^2(p_1, p_2) = \frac{1}{2} \sum_{i=1}^{h} (\frac{1}{\sigma_1^{i\,2}} + \frac{1}{\sigma_2^{i\,2}})(\mu_1^i - \mu_2^i)^2 \tag{6}$$

To learn and measure semantic similarity with our *variational siamese* network, we express the previous metrics "element-wise" and feed the resulting tensor as input to a Multi-Layer Perceptron $\psi$ that predicts the degree of similarity of the corresponding pair. By "element-wise" Wasserstein-2 tensor, we mean the tensor of dimension h whose $i$th element is computed as $(\mu_1^i - \mu_2^i)^2 + (\sigma_1^i - \sigma_2^i)^2 \in \mathbb{R}$, for a pair of vectors $(\mu_1, \sigma_1), (\mu_2, \sigma_2)$. The Wasserstein distance (scalar) is obtained by summing the components of this tensor. Our neural architecture is illustrated in Figure 4.

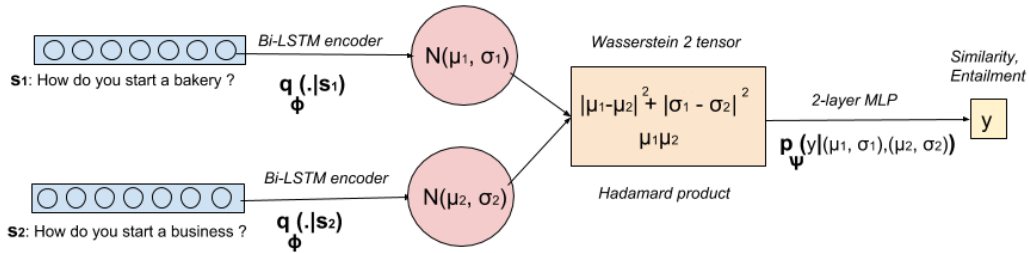

Figure 4: Our *variational siamese* network to measure and learn semantic similarity.

## 4.2 Neural architecture

Let $s_1, s_2$ be a pair of sentences and y a label indicating their degree of similarity (or entailment). Denote by $(\mu_1, \sigma_1), (\mu_2, \sigma_2)$ the inferred pair of latent intents. Our goal is to predict $y$ given $(\mu_1, \sigma_1)$, $(\mu_2, \sigma_2)$. Our pair of latent intents are used to compute the element-wise Wasserstein-2 tensor $(\mu_1 - \mu_2)^2 + (\sigma_1 - \sigma_2)^2 \in \mathbb{R}^h$ and the Hadamard product $\mu_1 \mu_2 \in \mathbb{R}^h$. The concatenation of the Wasserstein-2 and Hadamard tensors is fed to a two layer Multi Layer Perceptron $\psi$ with ReLu activations (inner layer) and softmax output layer for classification of the sentence pair $(s_1, s_2)$ as duplicate or not as shown in Figure 4. Our *variational siamese* network learns to detect paraphrases by minimizing the following regularized loss by stochastic gradient descent:

$$L_{\psi;\phi}(s_1, s_2) = -y \log p_\psi(y|q_\phi(.|s_1), q_\phi(.|s_2)) + \lambda ||\psi||_1 \tag{7}$$

The first term is the cross entropy of the prediction and the second term is a L1 regularization to encourage sparsity of our MLP's weights ($\lambda = 0.00001$). There is no $N(0, I)$ prior on intents during the training of our variational siamese network (VAR-siam) as observed in Figure 3.

# 5 Experiments and results

## 5.1 Experimental details

We implemented our model using python 3.5.4, tensorflow 1.3.0 [42], gensim 3.0.1 [43] and nltk 3.2.4 [44]. We evaluated our proposed framework on Quora question pairs dataset which consists of $404k$ sentence pairs annotated with a binary value that indicates whether a pair is duplicate (same intent) or not.[1]

**Data preprocessing**  We convert sentences to lower case, refit dashes for single words, space punctuation, currencies, arithmetic operations and any other non alphanumeric symbols. We tokenize sentences using nltk tokenizer [44]. We remove from our vocabulary words that appear at most once resulting in a total of 48096 non unique words. Unknown words are replaced by the token 'UNK'. We pad our sentences to 40 tokens (words).

**Neural architecture and training**  We embed words in a $300$ dimensional space using Glove vectors [6] pretrained on Wikipedia 2014 and Gigaword 5 as initialization. Our encoder and decoder share the same embedding layer. Our variational space $(\mu, \sigma)$ is of dimension $h = 1000$. Our bi-LSTM encoder network consists of a single layer of $2h$ neurons and our LSTM [31] decoder has a single layer with 1000 neurons. Our MLP's inner layer has 1000 neurons. All weights were randomly intialized with "Xavier" initializer [45].
We successively train our model on reformulating (for 5 epochs) and detecting paraphrases (3 epochs). The first task provides an initialization for the second one (embedding and encoding layer). For both tasks, we employ stochastic gradient descent with ADAM optimizer [46] ($lr = 0.001, \beta_1 = 0.9, \beta_2 = 0.999$) and batches of size 256 and 128. Our learning rate is initialized for both task to 0.001, decayed every 5000 step by a factor 0.96 with an exponential scheme. We clip the L2 norm of our gradients to 1.0 to avoid exploding gradients in deep neural networks.

## 5.2  Semantic question matching

The task of semantic question matching is to predict whether or not question pairs are duplicate. Table 1 compares different models for this task on the Quora dataset. Among models that represent the meaning of text pairs independently (siamese networks), our model performs strongly. More surprisingly, our model is competitive with state-of-the-art models that read sentences together before reducing them to vectors, such as Bilateral Multi Perspective Matching (BiMPM) [47], pt-DECATT [48] and Densely Interactive Inference Network (DIIN) [22].

Table 1:  Quora question pairs dataset result. The split considered is that of BiMPM [47]. Models are sorted by decreasing test accuracy. Results with † are reported in [22] and ‡ in [25]. SWEM [49] stands for Simple Word Embedding based Models.

| Model | Read pairs separately | Accuracy Dev | Accuracy Test | Generative (pre)training |
|---|---|---|---|---|
| DIIN [22] † | False | 89.44 | 89.06 | False |
| **VAR-Siamese (with repeat/reformulate)** | **True** | **89.05** | **88.86** | **True** |
| pt-DECATTchar [48] † | False | 88.89 | 88.40 | False |
| **VAR-Siamese (with repeat)** | **True** | **88.18** | **88.24** | **True** |
| BiMPM [47] † | False | 88.69 | 88.17 | False |
| pt-DECATTword [48] † | False | 88.44 | 87.54 | False |
| L.D.C † | False | - | 85.55 | False |
| Siamese-GRU Augmented [25] ‡ | True | - | 85.24 | False |
| Multi-Perspective-LSTM † | False | - | 83.21 | False |
| SWEM-concat [49] | True | - | 83.03 | False |
| SWEM-aver [49] | True | - | 82.68 | False |
| Siamese-LSTM † | True | - | 82.58 | False |
| SWEM-max [49] | True | - | 82.20 | False |
| Multi-Perspective CNN † | False | - | 81.38 | False |
| DeConv-LVM [37] | True | - | 80.40 | True |
| Siamese-CNN † | True | - | 79.60 | False |
| **VAR-Siamese (w/o pretraining)** | **True** | **62.16** | **62.48** | **False** |
| Bias in dataset (baseline) | - | 62.16 | 62.48 | - |

We considered two pretraining strategies: *repeat* (standard VAE) and *repeat, reformulate* (cf. section 3.). We achieved strong results with both settings (as shown in Table 1). Without a generative pretraining task, our variational siamese quickly classifies all pairs as non duplicate. Learning semantic similarity is a difficult task and our variational siamese network fails with randomly

initialized intents. Generative pretraining helps our variational siamese network learn semantic representations and semantic similarity.

## 5.3 Question retrieval

Siamese networks can read a text in isolation, and produce a vector representation for it. This allows us to encode once per query and compare the latent code to a set of references for question retrieval. For a given query, our model runs 3500+ comparisons per second on two Tesla K80.

When we combine our *variational siamese* network with Latent Dirichlet Allocation [50] to filter out topics, our approach is relevant and scalable for question retrieval in databases with a million entries. Related questions are retrieved by filtering topics (different topic implies different meaning) and are ranked for relevance with our *variational siamese* network. We report some human queries asked to our model on Quora question pairs dataset in Table 2.

Table 2: Question retrieval (best match, confidence) with our proposed model on Quora question pairs dataset (537 088 unique questions). All queries were retrieved in less than a second on two Tesla K80 GPU.

| Query | Retrieved question 1/537088 | MLP's output, Confidence |
| --- | --- | --- |
| Hi! If I run the speed of the light, what would the world look like? | What would happen if I travel with a speed of light? | 99.68% |
| How do you do a good gazpacho? | How can I make good gazpacho ? | 99.52% |
| What can I do to save our planet? | How can I save the planet? | 95.91% |
| What is the difference between a cat? | What is the benefit of a cat? | 90.99% |
| Can we trust data? | Can I trust google drive with personal data? | 57.13% |

## 5.4 Discussion

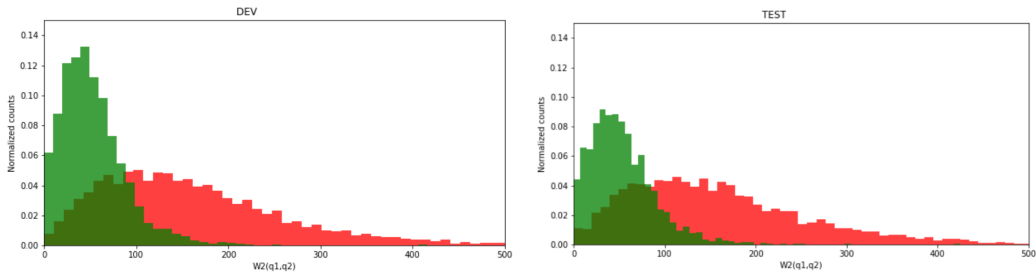

Figure 5: Empirical distribution of Wasserstein 2 distance between pairs of intents for duplicate (green) and not duplicate (red) pairs, after variational siamese training (with repeat, reformulate pretraining). Left: Quora Dev set. Right: Quora Test set.

As shown in figure 5, we measured the Wasserstein 2 distance on Quora dev and test set using different encoding (sentence representations) and reported AUC scores in Table 3. Our baseline is Word Mover's Distance [1] which considers documents as bag-of-words. We noticed that the standard VAE (repeat) underperforms the bag-of-word model. The unsupervised task of repeating sentences allows to encode information but not in a specific way (intent vs. style [38] eg.). Our proposed VAE (repeat, reformulate) performs better than our baseline. Our best results (2 first lines in Table 3) were obtained when further retraining a VAE discriminatively with our *variational siamese* network to learn a similarity metric in a latent space.

Intuitively, our model learns, with the Wasserstein-2 tensor, how to optimally transform a continuous distribution (intent) into another one to measure semantic similarity. We considered learning and

Table 3: Discriminative power of Wasserstein 1 and 2 on Quora dataset.

| Encoding | Wasserstein | Area under ROC curve | | Training |
|---|---|---|---|---|
| | | Dev set | Test set | labels |
| VAR-siamese (with repeat) | W2 | 87.11 | 86.74 | Positive |
| VAR-siamese (with repeat/reformulate) | W2 | 86.88 | 86.27 | & Negative |
| VAE (repeat/reformulate) | W2 | 77.70 | 77.44 | Positive |
| Word Mover's Distance [1] (bag-of-word) | W1 | 73.47 | 73.10 | None |
| VAE (repeat) | W2 | 70.67 | 69.91 | None |

measuring similarity with other metrics from Information Theory (Mahalanobis distance and Jensen-Shannon divergence), as input layer for our Multi-Layer Perceptron. All models overpassed $87.5\%$ dev accuracy on the Quora dataset (same split as in Table 1). As in [41], which addresses semantic image segmentation, we found Wasserstein-2 to be the most effective approach. Adding the Hadamard product to the MLP's input layer improves predictions by a marginal $1\%$. When trained with Wasserstein-2, the convergence is consistent across all tested similarity metrics (Mahalanobis, Jensen Shannon, Wasserstein 2, Euclidean, Cosine). This gives us hints about the informative nature of our learned latent semantic space.

# 6  Conclusion

Information Theory provides a sound framework to study semantics. Instead of capturing the relationships among multiple words and phrases in a single vector, we decompose the representation of a sentence in a mean vector and a diagonal covariance matrix to account for uncertainty and ambiguity in language. By learning to *repeat, reformulate* with our generative framework, this factorization encodes semantic information in an explicit sentence representation for various downstream applications. Our novel approach to measure semantic similarity between pair of sentences is based on these continuous probabilistic representations. Our *variational siamese* network extends Word Mover's Distance [1] to continuous representation of sentences. Our approach performs strongly on Quora question pairs dataset and experiments show its effectiveness for question retrieval in knowledge databases with a million entries. Our code is made publicly available on github.[2]

In future work, we plan to train/test our model on other datasets, such as PARANMT-50M [51], a dataset of more than 50M English paraphrase pairs, and further propose a similar framework (*variational siamese*) for the task of natural language inference [20] [21] to predict if a sentence entails or contradicts another one. A possible generative framework, similar to *repeat, reformulate*, would be *repeat, drift*. Learning Gaussian representations with non diagonal covariance matrices is also a direction that could be investigated to capture complex patterns in natural language.

**Acknowledgments**

We would like to thank Ecole Polytechnique for financial support and Télécom Paris-Tech for GPU resources. We are grateful to Professor Chloé Clavel, Professor Gabriel Peyré, Constance Noziere and Paul Bertin for their critical reading of the paper. Special thanks go to Magdalena Fuentes for helping running the code on Telecom's clusters. We also thank Professor Francis Bach and Professor Guillaume Obozinski for their insightful course on probabilistic graphical models at ENS Cachan, as well as Professor Michalis Vazirgiannis for his course on Text Mining and NLP at Ecole Polytechnique. We also thank the reviewers for their valuable comments and feedback.

## Footnotes

[1]https://www.kaggle.com/quora/question-pairs-dataset

[2]`https://github.com/MichelDeudon/variational-siamese-network`

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
