[Reviews · NeurIPS 2018]

Reviewer 1



The authors propose to encode a sentence using (mean, variance). First training an autoencoder and then tuning it further using Wasserstein 2 distance functions on labeled sentence pairs. The results show competitive performance from the proposal. By the repeat/reformulate do the authors just mean decode ? Quality/Significance Among models that encode query/sentence independently, the proposal performing the best is not a breaking result. This is because, - The similarity function (2-layer MLP) is reasonably expressive and a high cpu operation, comparable to the cost of encoding the text at runtime. - Because the costs are comparable, I do not see the benefit of this method vs BiMPM, pt-DECATT and DIIN. This would've been an important result if the similarity function was really cheap, e.g. just a dot-product, as this enables the use of fast NN techniques for question retrieval. Questions 1. "We did not consider training our model jointly on both tasks,... " Did the authors try this ? If so what were the results like ? Also why did the authors not consider combining both the tasks into a single task ? 2. what was the performance of question retrieval of BiMPM, pt-DECATT and DIIN. It might be slower, but surely, it can be computed ? EDIT: After author's response, I'm okay with accepting the paper since the authors promised to remove the claim of state-of-art among siamese networks (this was mentioned 4 times and I assumed that was the core contribution). Siamese networks are really useful when (a) there is improved test performance over non-siamese networks because of parameter sharing or (b) enable fast retrieval techniques when the distance metric between encodings are simple to compute. The proposed work does not show (a) and the distance metric is certainly not (b). Therefore the statement “state-of-art among siamese networks” is not very relevant since the model is not really giving any benefits of being a siamese network.

Reviewer 2



This work proposes a deep generative model for learning semantic representations of text. They propose two models, the first can be trained on raw text and is used to initialize the second for which supervision can be used. The paper is a extension of previous work where their sentence representations are Gaussians and not just vectors. They evaluate on the Quora dataset and do achieve "state-of-the-art" results, though they do overstate these. They achieve these only when sentences are modeled separately. Also this dataset is not as widely used evaluating sentence similarity - related work often evaluates on the STS tasks, though this dataset is another nice addition to evaluations of semantic similarity. This is also a source of missed references/comparisons in the work (see "Towards universal paraphrastic sentence embeddings" and "Pushing the Limits of Paraphrastic Sentence Embeddings with Millions of Machine Translations"). Therefore it is hard to see how strong their numbers are in comparison to other methods that also use pre-taining and learn representations instead of just a scoring function. Overall, I like the paper, it is difficult to get these VAE methods to work and it is a contribution to see how strong they can perform on these tasks given they have some advantages (like an interpretation) over earlier methods. I would've liked stronger comparisons and more evaluations (to be comparable with related work) - it would strengthen the paper a lot if they were competitive or better. It would also be nice to see how their unsupervised approach works on its own and how their results are better than other VAE methods that do not represent sentences as Gaussian distributions.

Reviewer 3



The paper addresses the problem of learning the representation of questions/sentences as an input unit. Like word2gauss, the embedding is into distributions (as opposed to a single point of a vector space) and in particular into Gaussian distributions. Embedding into a distribution enables the use of a transport distance, such as W2 for computing discrepancy between distributions which has advantages over alternatives such as KL divergence but is expensive to compute. Here the specific constraints on the distributions made the computation of the transport distance efficient. The application considered is question de-duplication. The authors solve the problem in two separate stages. The first stage is the actual representation learning stage, which embeds questions into the Gaussian distribution. The second stage is to learn from labeled data to decide if the two pre-trained representations correspond to equivalent questions or not. For the first stage a generative (LSTM based) VAE model is proposed to convert a question into itself or a similar question. For the second stage a Siamese structure is proposed that is based on W2 distances. The model is evaluated on Quora question deduplication dataset and performed strongly. I liked this paper: there is a creative component in the overall proposed solution and the architecture and the pieces fit together well. The embedding of a whole sentence to a distribution seems very natural, W2 seems like a good way to compute distances, etc. I also liked the idea of "similar-encoder", which could also be useful as opposed to "auto-encoder", as an augmentation method. The two stage development of the solution raises my curiosity though. The first stage, i.e. text generation, is a difficult task. Eliminating this stage completely and performing an end-to-end training would have been an alternative way to go. The pre-training from the first stage could still be used for initialization, but the question representations could have been fine-tuned for the actual task. I was hoping to see more discussions/results on that alternative. Moreover, it is surprising that no generation result is reported. The paper is well organized and readable. But it needs proofreading and improvement in presentation in places. Mathematical notations should be more carefully used. - The notations for vectors and scalars seem to be mixed up. See for example Section 3.2. There seem to be a dimension mismatch in Equations (3) and (4). The notation for matrix square root is unclear. - Section 4.2 is not clear. What is "element-wise" Wasserstein? Is it not that Wasserstein distance returns a number (scalar) for a pair of vectors? Is the dimension of the input to the next module h+1? (One for the W norm and h for the Hadamard product?) - Not enough details is given about the W2 computation for Gaussians. In particular, the first paragraph of page 6 (in Section 4.1) lacks enough details and is cryptic. Needs expansion. - Minor: "Similarly to","Sigm", "sentences representation", "euclidean", "gaussian", "figure 3", "[3][4][5][6] and [7]", "87,5". The approach proposed is novel to my knowledge. The work is a good continuation of the line of research that focuses on embedding into distributions. It is also a good application of W2 distance between distributions. The generation part of the proposal ( "Repeat-Reformulate") reminds me of "Sequence to better sequence" in [1]. Citation might be a good idea. The significance of the method is enough to be accepted I think. Learning representation for a question/sentence is an important and unsolved problem. Embedding to the right mathematical object is an important question and explorations and insights on that seems useful. Coming up with a robust distance between distributions that is easy to compute is also beneficial. Finally, the application considered has received attention lately in ML and NLP community. Overall, considering strengths and weaknesses of the paper, I vote for the paper to be accepted. ---------------------------- [1] Mueller, J., Gifford, D., Jaakkola, T. "Sequence to better sequence: continuous revision of combinatorial structures", in ICML 2017. ---------------------------- Thanks for the clarification on the "element-wise" Wasserstein operation that is mentioned. It makes more sense now. That should definitely be clarified in the write up. (Mathematically defining the ith element of the operation by using indices would be helpful to clarify the definition.)